# Out of Sight: Social Control and the Regulation of Public Space in Manchester

Christopher J. Moss [1] and Kate Moss [2,*]

[1]  School of Environment Education and Development, University of Manchester, Oxford Road,
    Manchester M13 9PL, UK; christopher.moss-8@postgrad.manchester.ac.uk
[2]  School of Law, University of Wolverhampton, Wolverhampton WV1 1AD, UK
[*]  Correspondence: k.moss@wlv.ac.uk

**Abstract:** This paper considers the history and context of the control of public spaces, how this is regulated currently and how it relates to the politics of homelessness and community governance with a specific focus on the regulation of public space in the contemporary city of Manchester.

**Keywords:** public spaces; public policy; homelessness; regulation; criminalisation

## 1. Introduction

In March 2019, figures from the Ministry of Justice reported that between 2014 and 2017, 6518 people were found guilty—under the Vagrancy Act 1824—of rough sleeping. Of these, 441 rough sleepers were charged and convicted of this offence in Greater Manchester alone. (Manchester Liberal Democrats Press Office 2019). In 2017, 21 rough sleepers died on the streets of Manchester (Booth 2019). This represents the highest number of deaths for any city in the United Kingdom; the national total for this in 2017 was 600. Manchester City Council is currently carrying out a public consultation on the implementation of new public space protection orders (PSPOs) covering four specific areas—Piccadilly Gardens, Chinatown, Piccadilly, and the Smithfield Estate in the Northern Quarter (Manchester City Council 2019). The council has indicated that this is a response to rises in complaints from city centre residents and businesses about anti-social behaviour and drug taking in the city centre. Labelled by the Liberal Democrats as a "Homeless Tax" (rough sleepers sheltering in doorways, beggars, and drug takers could be given on the spot fines of £100 which, if unpaid could lead to imprisonment) this approach has been criticised as an "out of sight out of mind" approach that criminalises some of the most vulnerable people in society and arguably represents a poor example of public policy because it does nothing to tackle the essential problem of rough sleeping and homelessness. Whilst Greater Manchester is not alone in adopting this sort of approach, for the purposes of this paper the focus will be on this region as an indicator of possible responses to a wider social issue and on the specifics of how the regulation of public space can arguably be used to cleanse public spaces of those individuals in society whose presence is deemed to be a nuisance.

According to Rubin (2018, p. 5) public spaces are either any area where individuals can interact with each other or spaces that have their use "dictated by the social contract of governance and the people"—such as parks, pavements, highways, squares, and some public buildings. These spaces have traditionally had a range of uses. Public spaces and their use are controlled in various ways for various reasons. Controlling the use of space and the issue of who is permitted to use public space, when and how, has its origins in the California School of Urbanism and ideas of defensible space. Recently, this has been contributed to in the United Kingdom by academics such as Armitage and Ekblom (2019) whose seminal work on Crime Prevention through Environmental Design (CPTED) has made an important contribution to a practice-based approach to reducing the risk of certain offences being

committed by the modification of the built environment. However, this perspective has also attracted some criticism for failing to integrate with other approaches, and whilst Armitage and Ekblom have responded to this, it remains the case that greater regulation of public space has both negative and positive aspects. Whilst acknowledging the importance and rigour of CPTED and related techniques, we want to talk in this paper about the development and current application of spatial regulation in the contemporary city of Manchester, whether this represents the neoliberalisation of society and space (Allmendinger 2017), and how this is currently affecting the homeless population of the city.[1]

## 2. The Regulation of Public Space in Contemporary Manchester

At the time of writing, Manchester City Council is proposing new powers to address a number of anti-social behaviours including obstruction of the highway, the obstruction of building entrances, stairwells, and exits, and occupying a tent or other temporary structure in a manner which is likely to cause a health and safety risk. These powers would take the form of new public space protection orders (PSPOs) which if breached, would authorise a council officer or police officer to issue an on the spot fixed penalty fine of £100. If this is not paid individuals can be prosecuted, fined up to £1000, or imprisoned. It is worth pointing out that Legal Aid is not an option for those wishing to defend themselves. Whilst the council has said that these restrictions would not be targeted at specific groups of people, it is difficult to imagine that the new powers we have identified above would not be used specifically in relation to the homeless in city centre Manchester. Additionally, there are already statutory powers (such as Obstruction of the Highway under s137 Highways Act 1980 and various other offences contained in the Anti Social Behaviour Crime and Policing Act 2014) covering these issues, so why would Manchester City Council wish to replicate measures that are already in existence?

One reason might be that using a PSPO makes it easier to punish someone for perpetrating the identified behaviours. Issuing a PSPO is immediate and negates the necessity of bringing charges against an individual and taking them to court. This in turn arguably denies that individual the right to a fair hearing under the Human Rights Act 1998 Article 6. Additionally, this approach also gives officers of the council and the police a great deal more flexibility in dealing with individuals they deem to be in breach of PSPOs and imparts an element of subjectivity in deciding whether certain behaviours amount to a breach. This is worrying since the types of people that will doubtless be targeted by these new measures will be some of the most vulnerable in society. It allows the Council (composed essentially of politicians for whom being seen to be tackling the problem of homelessness is a huge political issue) the ability to punish people for their behaviour without having to refer them for prosecution. It simultaneously puts any council officer who sees a person in a doorway/stairwell/exit in an invidious position. On the one hand, do they seek to punish this person by issuing them with a fine for contravening the new PSPO or do they—believing this person to be homeless—seek to help by responding to their legal duty to provide suitable accommodation for that person? Putting such decisions in the hands of politicians is surely poor public policy, not least because no safeguards will exist to ensure that these additional powers are not misused.

There is perhaps another aspect to this that may be more covert. The council would doubtless argue that these measures are not specifically targeted at the homeless (because they do not specifically mention homeless or destitute people) and as such comply with Home Office Guidance issued in 2018 (Home Office 2018) which instructed councils not to target people for being homeless and sleeping rough. However, this would necessitate the council arguing that some people who sleep in doorways or keep their belonging there, are not actually homeless or destitute. The end result of course will still be the same; that the homeless and destitute will be disproportionately affected by these PSPOs. Rough sleepers and the homeless stand very little chance of avoiding behaving in the ways outlined by

---

[1]  It is not our intention to use this article to discuss the diversity of the homeless population, which has been discussed inter alia by ODPM (2005); Dwyer and Brown (2008); and Fitzpatrick et al. (2011).

Manchester's proposed new PSPOs. They will thus find themselves in a more hostile environment, will quickly be dealt with by way of fines, and will ultimately be pushed out to the farthest reaches of the city where ironically they will fall outside of the remit of the Council's Homeless Advice and Assessment service, which has a statutory duty to help them. It is pertinent to address the historical and current mechanisms of legal spatial regulation that facilitate these approaches to homeless populations in Manchester and elsewhere.

## 3. The Legal Context of Spatial Regulation

Currently still in force in the United Kingdom is the Georgian Vagrancy Act 1824, which was specifically targeted towards preventing soldiers, returning from the Napoleonic War, from sleeping on the streets of London. The history of this goes back even further to the 16th century when anti-vagrancy measures were introduced to tackle an increase in the numbers of homeless following the dissolution of the monasteries and Elizabethan legislation against beggars, suspected witches, and gypsies. The forerunner of modern vagrancy laws was the Vagrant Act 1744, which divided beggars and so-called "idle persons" into categories such as the unemployed, those refusing to work, those not supporting their families, rogues and vagabonds, and "incorrigible" rogues who had been convicted of an offence or offences previously. These wide definitions enabled the authorities to operate a policy of clearance of virtually any person deemed to be "not giving a good account of themselves". Negative attitudes towards these categories of people were compounded by the introduction, in the 18th century, of rewards for identifying vagrants, beggars, or "disorderly people". Inevitably this practice was open to abuse and moral panics about vagrants turning into more serious criminals such as burglars and highwaymen. The Vagrancy Act 1824—which is still enforceable today—came about as a result of such fears and its wide remit can be noted from sections such as 1 and 4;

Section 1:

Every person wandering abroad, or placing himself or herself in a public place, street or highway, court or passage to beg or gather alms, or causing or procuring or encouraging any child or children so to do, shall be deemed an idle and disorderly person. On conviction following the evidence of one or more credible witness or witnesses such an offender could be jailed for one month.

Section 4:

Every person pretending or professing to tell fortunes, or using any subtle craft, means, or device, by palmistry or otherwise, to deceive and impose on any of his Majesty's subjects;

Every person wandering abroad and lodging in any barn or outhouse, or in any deserted or unoccupied building, or in the open air, or under a tent, or in any cart or wagon not having any visible means of subsistence and not giving a good account of himself or herself;

Every person wilfully exposing to view, in any street, road, highway, or public place, any obscene print, picture, or other indecent exhibition;

Every person wilfully openly, lewdly, and obscenely exposing his person with intent to insult any woman;

Every person wandering abroad, and endeavouring by the exposure of wounds or deformities to obtain or gather alms;

Every person going about as a gatherer or collector of alms, or endeavouring to procure charitable contributions of any nature or kind, under any false or fraudulent pretence;

Every person apprehended as an idle and disorderly person, and violently resisting any constable, or other peace officer so apprehending him or her.

This antiquated legislation is still used to move on the homeless, particularly if a police officer thinks that person has been begging, and fines can also be imposed. It is somewhat futile to fine a homeless person, who would not have the money to pay a fine and this approach does nothing to solve the underlying problem of homelessness. This type of approach has been summed up by Monbiot

(2014) who comments that "until the late 19th Century, much of our city space was owned by private landlords. Squares were gated, turnpikes controlled streets. The great unwashed, many of whom had been expelled from the countryside by acts of enclosure, were also excluded from desirable parts of town. Social reformers and democratic movements tore down the barriers, and public space became a right, not a privilege. But social exclusion follows inequality as night follows day, and now, with little public debate, our city centres are again being privatised or semi-privatised. They are being turned by the companies that run them into soulless, cheerless, pasteurised piazzas, in which plastic policemen harry anyone loitering without intent to shop. Street life in these places is reduced to a trance-world of consumerism, of conformity and atomisation, in which nothing unpredictable or disconcerting happens, a world made safe for selling mountains of pointless junk to tranquilised shoppers. Spontaneous gatherings of any other kind—unruly, exuberant, open-ended, oppositional—are banned. Young, homeless and eccentric people are, in the eyes of those upholding this dead-eyed, sanitised version of public order, guilty until proven innocent".

Provisions in the Anti-Social Crime and Policing Act 2014 (ASCPA) now include powers to ban certain activities from designated areas using Public Space Protection Orders (PSPOs). It has long been argued that existing laws to regulate the use of public space—such as those which are still enforceable under the Vagrancy Acts—are over-used, too broad, and have been employed unjustly to interfere with law-abiding individuals. The danger is that PSPO powers reinforce a heavy handed approach to those who live on the streets. They also have the potential to be abused since they require less public consultation than those relating to alcohol or dog-control areas. At the moment, PSPOs can be used by councils to ban homeless people from begging or sleeping in public places such as parks. The provisions can be directed at particular groups, and thus there is potential for them to be used disproportionately by race or gender. Homeless people already struggle to assert any form of stewardship over the public spaces they have to occupy. This continues to be an under-explored area of socio-cultural research. The stereotyping of homeless people as dangerous and generally untrustworthy with regard to their use of public space, has already led to national and local policies that enforce laws that specifically criminalise homelessness. Laws which are used against the homeless, such as the Vagrancy Act 1824 and the ASCPA 2014 are thus seen as "quick-fix" solutions—removing homeless people from sight. However they fail to tackle the root causes of homelessness or provide any remedial attention to the issue of homelessness.

It is difficult to envisage any sort of behaviour that could potentially not be classed as a "nuisance" or "annoyance" to someone, at some time. People often complain about young people who tend to hang around outside, saying that they appear aggressive and intimidating. The breadth of the concept of nuisance or annoyance is such that it could arguably be applied to anything and this could have far reaching and serious implications for society—buskers, orators, preachers, street artists—and for free speech. Arguably the ASPCA 2014 is one of the most oppressive pieces of legislation any recent Parliament has enacted—and Monbiot (2014) makes a good point when he says that "the new injunctions and the new dispersal orders create a system in which the authorities can prevent anyone from doing more or less anything [ . . . ] These laws will be used to stamp out plurality and difference, to douse the exuberance of youth, to pursue children for the crime of being young and together in a public place, to help turn this nation into a money-making monoculture, controlled, homogenized, lifeless, strifeless and bland".

## 4. The History and Context of Spatial Regulation in Manchester

For the purposes of this paper, and as a result of the current interest by Manchester City Council in extending the remit of is spatial regulation in the form of eight new public space protection orders, it is relevant to discuss the background and historical context of spatial regulation within Manchester. From the 1700s the industrial city emerged; spatially fixed, with the ability to grow and develop through means of economic production. Since the 1930s, cities have again undergone transformation, characterized in part by periods of deindustrialization and the rise of service industries

(Savage and Warde 1993). For Harvey (1989, p. 124), 1973 marked the turning point from a Fordist system of mass production and consumption—which had defined the post-war era—to a post-Fordist or post-modern, "flexible" regime of accumulation. The emergence of postmodernism within this discourse of capital accumulation, labour markets and organisations has shaped both the physical and cultural development of cities and societies around the world. Famously, Foucault (1970, 1977) viewed the forces that have forged the contemporary postmodern society critically, rejecting established notions of freedom and economic and social liberation. Instead, he questioned the origins of power, claiming forces of exclusion, control and violence constituted a cultural hegemony at work, defined by capitalism. It is the consequences of this new era of economic activity; the socio-spatial impacts of postmodern urban development that have been the subject of much theorising by the California School of Urbanism. In this view, the postmodern urban environment has become a fortress, where interdictory forces work to maintain a particular mode of socio-spatial regulation, from which those within it cannot deviate (Davis 1990; Flusty 1994; Soja 2000).

Manchester experienced great industrialisation and rapid deindustrialisation, giving way to periods of urban decay, regeneration and extensive development. This "urban renaissance" is symbolic of the government's ambition since the 1970s and typifies the "function of international financial speculation on an unprecedented scale" (Davis 1985, p. 109). Whilst referring to Los Angeles here, comparisons can be drawn with Manchester given the global "rise of neoconservatism and the privatisation ethos" which have been particularly influential in both the United States and the United Kingdom (Dear and Flusty 1998, p. 58). The adoption of neoliberal political agendas has changed the nature of cities, "*how* they operate circuits of power", and the way that this in turn creates, reinforces and reproduces socio-spatial inequalities within them (Coleman 2004, p. 84, original emphasis). Such circuits seek to establish and maintain a particular social order, that ultimately is constructed around a specific set of inherently capitalist ideals and values (Massey 2011), which are then typically interpreted as those of the state. What the California School began to identify are the ways in which such control is exercised throughout the city by various regulatory processes. Of the 10 categories that constitute Dear and Flusty's (1998, p. 54) "taxonomy of Southern California Urbanisms"[2], four are present in Manchester's Business Improvement District (BID)[3], the epitome of the "fortified city", a network of "interdictory space[s]", where the processes of both Fordist and Post-Fordist regimes of accumulation, regulation, and globalization contribute to maintaining a specific exclusionary milieu of consumerist conformity.

Through a combination of planning strategies, architecture, design, surveillance, security and policing the urban environment begins to resemble the fortress that Davis (1990) envisioned. Social anxieties are produced and reproduced, exacerbating fears of the public sphere and perpetuating the "naturalization of interdictory space" (Flusty 2001, p. 661). The quintessence of a neoliberal planning agenda in Manchester is the development of the BID, the culmination of over a decade of private sector decision-making in the city. Since 2000, Cityco has run the centre of Manchester, a private company funded by local businesses and chaired by the main property developers in the city (Minton 2012). In 2014, Cityco established Manchester's first BID stretching from the top of Market Street down to Deansgate, incorporating the Arndale Centre to the North and King Street to the South (Cityco 2013, pp. 14–15). The role of the BID is to enhance both the city's image (on the regional, national and international stage) and the retail conditions of those businesses that can afford membership. However,

---

[2]  Flusty's (1994, pp. 16–17) taxonomy of interdictory spaces present in the fortress city: "Some spaces are passively aggressive: space concealed by intervening objects or grade changes is 'stealthy'; space that may be reached only by means of interrupted or obfuscated approaches is 'slippery'. Other spatial configurations are more assertively confrontational: deliberately obstructed 'crusty' space surrounded by walls and checkpoints; inhospitable 'prickly' spaces featuring unsittable benches in areas devoid of shade; or 'jittery' space ostentatiously saturated with surveillance devices".

[3]  Manchester BID is a consortium of more than 400 leading retailers and restaurants whose remit is to "invest to enhance Manchester's central shopping area". More details can be accessed at https://cityco.com/about-manchester-bid/ (accessed 5 April 2019).

this shift in power marks the establishment of private modes of governance and decision-making that reduces local democracy and arguably represents "a new culture of authoritarianism and control" (Minton 2012, p. 40). Although privatization is not a new phenomenon and has been a feature of the U.K. government's agenda since the Thatcher administration, the power granted to organizations through the recent establishment of BIDs is unparalleled.

Within the boundaries of Manchester's BID all elements of Flusty's (1994, pp. 16–17) interdictory space are experienced in some form, but perhaps most prominent is the extensive network of "jittery" space, saturated with security and surveillance devices. Soja (1989, p. 237) has used the term "urban panopticon" to describe the increasingly intrusive means of securitization that are now well established in the contemporary city of the 21st century. The idea draws on Foucault's (1980) theorizing around Jeremy Bentham's design of the panopticon prison, whereby power, influence and control is exercised over the subject through the perception of total and constant surveillance. The exertion of power through varying configurations of space to instil a particular urban milieu is an integral part of the California School of Urbanism's theories. Countless surveillance cameras, private security personnel and police work to create spaces that "extirpate the spontaneous, the unpredictable, free expression, dissidents, alien cultural practices and the insufficiently affluent from the built environment" (Flusty 2001, p. 661). Nowhere is this more apparent than within BIDs, where the excitement traditionally associated with the capricious nature of cities is replaced with sanitized, planned, almost clinical environments, which "internalize control, morals and values", disciplining the body to act *correctly* when traversing the new urban environment (Galič et al. 2017, p. 16). However, such extreme interpretations are not without their critics and Davis' (1990) hardline approach to socio-spatial regulation has even been argued by (Soja 2000, p. 323) to be a "tightly focused political philosophy of angry Marxian anti-neoliberalism". Though militant and totalizing in his delivery, Davis' (1990) account of Los Angeles is increasingly relevant when attempting to understand the myriad regulatory forces foisted on contemporary urban environments since the turn of the century.

Surveillance and spatial control have become an integral component of the means by which the urban environment is governed. Foucault (1980, p. 39) viewed this as active Panopticism "embedded in the very nature of modern life, so much so that we cease to recognize its existence". This process of "normation" (Foucault 1973), by which certain sets of attitudes, behaviours and habits become institutionalized creates new, accepted ways of doing, being, thinking and living. The importance of this within the contemporary city is that a consumer-focused culture becomes centralized, which produces and reproduces socio-spatial inequalities. Here, citizens are increasingly viewed as consumers "seduced into the market economy" and those who fall outside this category become increasingly marginalized (Galič et al. 2017, p. 22). This "embeddedness" of inequality in urban restructuring processes is described by Soja (2000, p. 267) as a "counter-discourse that insistently normalizes social inequality and represents it as an intrinsic part of all contemporary capitalist societies". The means by which this is achieved, at least in part, can be attributed to the reproduction of collective anxieties and the demonization of certain social groups. The opening gambit for Manchester's 2014 BID plan starts: "Manchester city centre is a world-famous destination for many reasons, but it's by no means secure in its position. We live in an uncertain world, where the most unexpected events can throw everything off balance [ . . . ] Take 2011; the riots that started in Tottenham spread rapidly to the rest of the UK. Manchester city centre was not spared" (Cityco 2013, p. 3).

It is this type of marketing and representation of space that creates and reinforces social anxieties, increasing divisions and levels of inequality throughout the city. It is also an attempt to justify the creation of a BID, intimating greater levels of security and protection are ensured with its establishment of a Business Crime Reduction Partnership (BCRP), working closely with businesses' security intelligence and the Greater Manchester Police (Cityco 2016). The crescive privatization and securitization associated with the development of the BID seeks to achieve Cityco's (2017, p. 5) vision of "a safe, clean and high quality city centre full of life, excitement, dynamism and, of course, customers". Minton (2012, p. 45) argues that "safe and clean" is much more concerned with the construction of

spaces that are "for certain types of people and certain activities" than they are with actual safety. Alluding to the interdictory nature of these spaces once again it appears as though cities are becoming entirely neoliberalised; products of privatization and capitalist economics that are primarily governed *by* and *for* markets (Akers 2015, p. 1843).

The intent, purpose or agenda of urban environments from a California School of Urbanism perspective is to regulate. "Market driven" and "class orchestrated" forces create "carceral structures", which represent the "archisemiotics of class war" (Davis 1990, p. 256). Just as surveillance is influential in the creation of a particular milieu, so too is the physical nature of space. For Gold and Revill (2003, pp. 36–37) landscape "naturalizes in material form the values of the powerful marking out moral geographies that exclude and exile feared social groups". Manchester's BID could be described—even after three decades—as a fitting representation of what Relph (1987) calls "quaintspace"; that which masks its regulatory nature under an aesthetically pleasing veil. This is the product of "a pretty lie" told by governments and corporations, which inconspicuously manipulates space in the pursuit of greater rationality and efficiency (Relph 1987, p. 259; Dear 2000). The purpose of the BID is to optimize the consumer experience, targeting "more high end, affluent shoppers" to increase the profitability of its 380 member businesses (Cityco 2013, p. 11). What this entails is a greater commodification and securitization of space, simultaneously reducing public freedoms and stymieing those vexatious behaviours—such as rough sleeping—that are not conducive to profit maximization.

One particularly disconcerting issue with this privatization of space and the creation of BIDs is the extent to which civil rights are restricted within them. Minton (2012) argues that such spaces are the consequence of specific policies towards planning and development that demonstrate the dominance of a neoliberal model of governance that has emerged since the 1980s. "Secured by Design", the U.K. Police's flagship crime prevention initiative exemplifies the strategies that characterize the fortress city, where the undesirable "debris" of the neoliberal environment—crime, litter, graffiti, the homeless, groups of teens, prohibited street traders—can be designed out of space (Minton 2012; Coleman 2004, p. 70). Spaces such as Market Street and Exchange Square in Manchester city centre are made inhospitable and uncomfortable, as well as impenetrable, with a distinct lack of places to sit, rest or linger for long periods. What (Flusty 1994, pp. 16–17) defined as "prickly" space has come to characterize much of the contemporary urban environment within the BID, where the homeless in particular have no place. The prolificacy of crime control and prevention measures in U.K. cities reflects wider anxieties towards difference that are reproduced within a neoliberal social order; where a citizens' value is determined by their ability to contribute to the economy through consumption.

Although somewhat unyielding in its approach, much of the California School of Urbanism's work is crucially important when interpreting the regulatory mechanisms that operate in contemporary Manchester. The representation of BIDs as a panacea for the socio-economic complexities of the urban environment is troubling. The entrenchment of a series of neoliberal planning and development strategies combined with a continuing retraction of state welfare support provides ideal opportunities for corporations to expand their sphere of influence. From a California School perspective, this new social order produces carceral spaces of capital accumulation, where the regulatory forces of interdiction exert power over citizens, normalizing processes of social peripheralization, which attempt to marginalize the insufficiently affluent from the built environment (Davis 1990; Dear and Flusty 1998; Flusty 1994; Flusty 2001; Soja 1989; Soja 2000). Though this appears to render citizens as completely passive to forces of regulation, when drawing comparisons to Engel's Manchester of the 1800s, Soja (2000, p. 322) argues that such "apartness also created a new synekism of identity, resistance and struggle". Therefore, the contemporary city is not completely bereft of resistance to the advanced forces of regulation that are imposed upon it, however the "universalization [ . . . ] of capitalism itself [ . . . ] the logic of capital accumulation and profit maximization" (Wood 1997, p. 551) that has come to penetrate almost every aspect of modern life makes acknowledging such forces increasingly complex.

## 5. Discussion

What we have outlined here are what we see as some of the reasons for the historical and contemporary focus on spatial regulation both generally in the United Kingdom and more specifically within the city of Manchester. Whilst there are sometimes good reasons for the control of public spaces—which academics focusing on the control and management of the built environment and attendant measures to reduce crime through environmental design provide important and informative discourses on—our interest is specifically on some of the more negative aspects of this type of control and how this specifically affects some of the most vulnerable people in society currently; the homeless and those who find themselves sleeping rough. It seems to us important to provide an alternative account of this not least because practices of controlling public spaces—for all of the reasons we have outlined—can have severe (and often unrecorded) consequences for the homeless. The regulation of public space restricts the life spaces of homeless people in the main because it deprives marginal groups who spend most of their day in public space or locations where congregation for social interaction—or at least some degree of personal comfort in keeping warm and dry—is critical (Doherty et al. 2008, p. 12). The reduction in the life spaces of the street homeless is the most direct evidence and the most obvious indicator of how the regulation of public space impacts on the life of the homeless. Such control of public spaces indicates a profound change in the social construction of homelessness, which can have serious consequences on policies. Flusty (1994) argues that this approach results in the homeless having no place, due to the inherent design of space. Within the contemporary landscape of the built environment this, plus the implementation of new criminalising orders such as PSPOs and the continued application of archaic vagrancy legislation results in the homeless being seen as having no value to the neoliberal social order where planning and development strategies are implemented through a combination of both design and marginalization strategies. Flusty (2001, p. 660) describes this process as "an ongoing dual process functioning to render ever higher levels of surveillance and social control and their recipients' corollary social peripheralization, publicly acceptable".

It appears that the social problem of homelessness is being framed almost exclusively in terms of public order or social nuisance problems and this has the effect of negating the search for solutions to homelessness as a worthy social issue. It then becomes a problem of negative behaviour which should be restricted and oppressed, rather than one attracting the application of positive social policies and—according to (Tosi 2007, p. 226) "reflects an individualist/social pathology perspective which seeks to make homeless people responsible and even guilty for their own situation. By de-socialising the problem and reducing it to a principle of 'order' it attempts to eliminate homelessness literally by directing effort towards making homeless people invisible, rather than meeting their needs". What this commentary about Manchester demonstrates is a reliance on policies of control, which can be explained in part by the social construction of homelessness, which Sahlin (2006) explains as a "struggle about definitions—of people, places and acts". Within this construction the homeless are criminalised by the phenomenon of control and a process of zero tolerance, facilitated by laws which are applied with the specific purpose of controlling public space. The application of zero tolerance theory thus becomes a discourse mechanism in which certain social behavioural patterns become connected with antisocial behaviour and criminality linked specifically to the homeless.

Discourses about homelessness should not be based on these associations but there is clearly confusion about this issue, which is summed up by Kelling and Coles (1996, p. 67) who comment that "advocates should preserve the myth that every person who begs aggressively, who lives in an encampment in a city park, or who urinates, defecates, or engages in sexual acts in public, is homeless. After all, making the problem of homelessness as vast as possible lends a compelling urgency to their argument". More recently both academic and political discourses have framed this issue in relation to ideas of "active" and "passive" citizenship (Levitas 2005; Tonkens and Doorn 2001). Whiteford (2008, p. 88) has suggested that "the cultural logic of this civic stratification of citizenship leads inexorably towards an overarching focus on responsibility and community processes which privilege individual duty and autonomy, communitarianism and a neighbourhood level-focus on the social and

cultural as well as economic dynamics of exclusion". Within this authoritarian process, homelessness is portrayed as a result of personal failure in an unequal society increasingly hostile to those who do not, or indeed cannot, confirm to social norms and where citizenship draws increasingly on the lexicon of obligations rather than rights (Roche 1992). Whiteford (2008, p. 89) sees this as an example of neo-liberal order, which "turns away from emancipatory and egalitarian goals associated with traditional welfare paternalism [and in which] rough sleeping has become the iconic subject of social exclusion".

The criminalisation of the homeless via the regulation of public space not only represents poor public policy but also shifts the burden of this social problem into the criminal justice domain. Shuffling off responsibility for this complex social problem to criminal justice agencies is particularly inappropriate in situations where the related issues—such as mental illness and substance abuse—that homeless people can often experience, should be handled by skilled service providers, not by criminal justice personnel who are not trained to respond to the situations that arise and cannot provide the necessary treatment and rehabilitation opportunities that may be required. Criminalisation thus provides no long-term benefit for homeless individuals nor does it provide a lasting solution to the conflicts over public space. Ultimately, the cycle of homelessness will only be broken when policies address the causes and effectively move people into housing. Currently it appears that there is a hostile policy environment characterised by punitive and exclusionary responses to homelessness, which serve only to regulate homeless populations. "[This] new paradigm subtracts the question of homelessness from integration policies: reduced to a principle of order, it is no longer a social welfare policy issue" Tosi (2007, p. 233).

The phenomenon of homelessness in contemporary western society is of course challenging. The ways in which the public space that the homeless occupy is regulated does nothing to further the policy or practice of social inclusion and how this is currently playing out in the contemporary city of Manchester demonstrates a number of things. Homelessness is often the end result of enormously complex social issues and homeless people are a diverse population whose social problems often stem from trauma. Often the only difference for those of us who suffer trauma and do not become homeless is usually the presence of social networks. Given this, it is all the more inappropriate that archaic vagrancy laws and public space protection orders are currently being used in a way that indicates that certain people in society are less worthy than others. With an absence of social networks and nowhere to go, it matters not how much you fine, imprison, or take someone to court; that will not prevent a person from being homeless. In this sense, Manchester is simply one example of what is happening to the homeless within some of the biggest cities of the United Kingdom. The continued victimisation of the homeless using both centuries old and modern laws will not get the homeless off the streets, will not provide adequate housing, and will not solve this tragic social problem; as such, it represents a very poor public policy response.

**Author Contributions:** Conceptualization; writing—original draft preparation; writing—review and editing all carried out by C.J.M. and K.M.

**Funding:** This research received no external funding.

**Conflicts of Interest:** The authors declare no conflict of interest.

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
