# Peer review of "Out of Sight: Social Control and the Regulation of Public Space in Manchester"

_socsci, doi:10.3390/socsci8050146_

Round 1
Reviewer 1 Report
This is a most interesting paper on the regulation of public space in Manchester. It discusses important historical and current legislation relevant to responding to visible homelessness in public spaces in the city. The paper provides excellent critical analysis about how the criminalising of homelessness contributes to increasing social inequalities/marginalisation and shapes constructions of homelessness as a 'public nuisance', which becomes prioritised in political discourse and policy making processes. It provides excellent theorising of surveillance and spatial control drawing on Foucault's notion of the active panoptican and resistance.I would recommend this paper for publication. it provides important critical commentary that is relevant to many cities beyond Manchester.
Author Response
Many thanks for your positive review.
Reviewer 2 Report
I read this piece with interest and pleasure. It is a good introduction to regulation of public space and it presents an interesting context of Manchester, establishment of BIDs etc. My main concern is however about the nature of this text. Is it a research article, a policy review or an essay? It seems to me, that it is foremost a commentary or a think piece. Which is fine by me, if that is also what editors of SS expect.
If so, it’s not a problem that it is not clear, what authors have done (Analysed historic and recent legislation? Made an inventory of recent renovation projects of public spaces in Manchester? Analysed data on fines and arrests by the police?). I generally agree that those changes lead to criminalisation of people living “on the streets”, but is there any evidence from Manchester that this regulation selectively affects people in homelessness? What about other policies to combat and/or prevent homelessness (temporary accommodation, social housing) in Manchester?
It’s an interesting text, but in my opinion this is an essay. It’s not a review or research article, it’s mostly presenting authors’ opinions about regulation of public spaces and effects it has on approaches to homelessness.
One thing I miss – from an essay type of text – is a broader British context of “hostile environment” towards migrants. Is there any evidence that these policies in Manchester are aimed not so much against (own) homeless, but against migrants?
Author Response
We would describe this article as a think piece / critical analysis / commentary which focuses on the criminalisation of homelessness and how that contributes to increasing social inequalities. We have framed this within theories of surveillance and spatial control and to do this we have drawn inter alia on Foucault's notion of resistance and the active Panoptican.
It is not the intention of the article to focus on homeless migrants per se. We have made this clear at the outset; that the intention was to raise issues about the construction of homelessness as a 'public nuisance', which becomes prioritised in political discourse and policy-making processes. The issue of homeless migrants – whilst obviously a salient one – is something that should be tackled in a stand-alone article, since the issues here are so complex that to include them here would merely pay them lip service. The homeless population is a diverse and complex one that this paper does not seek to comment on since that is a separate issue, which has been covered by academics numerous times. However, we have more clearly outlined this with a footnote in the introduction section of the paper and there is also already reference to this on page 9.(The revised manuscript can be forwarded at any time).
If we have understood the comment at the start of the second paragraph correctly – “If so, it’s not a problem that it is not clear, what authors have done?” - we don’t agree that the problem is necessarily that clear. The links between the imposition of new PSPO’s and the types of negative effects on the general homeless population in Manchester has not been raised before; neither has the theorizing as to exactly why this might be happening. As such we feel that this is a very relevant example of poor public policy making that should receive more academic attention. We don’t believe that these theoretical and policy explanations have been used before specifically in relation to what is currently happening in Manchester in relation to BIDs and it is all the more relevant given Manchester’s relatively poor statistics in relation to homelessness (which are referred to at the start of the article) and since the public consultation on the imposition of new PSPO’s has only just taken place.
Not all academic articles have to be research articles. There has always been a place within the peer-reviewed sphere for articles that could be termed ‘think pieces’ or commentaries. The value of these is to interrogate an idea, raise an academic discussion or to invite varied interpretations surrounding a construct or scholarly interpretation of a given paradigm. Articles such as this are intended to provoke ideas and discussion and should not be seen merely as narratives but as persuasive presentations of key concepts. We feel it is important to open up the debate about policy issues to a wider audience and to provide evidence that certain policies that are being adopted by local government are having a detrimental effect on communities of people that live within the city. We have also made it clear that it provides important critical commentary that is relevant to many cities beyond Manchester. However, we do not claim to have evidence or information from other cities – hence the fact that we have concentrated on the public policy processes currently in play within the city of Manchester and an explanation as to why that might be the case.